# The Ultimate List of the Most Frightening and Disgusting Animals: Negative Emotions Elicited by Animals in Central European Respondents

**DOI:** 10.3390/ani11030747

**Published:** 2021-03-09

**Authors:** Helena Staňková, Markéta Janovcová, Šárka Peléšková, Kristýna Sedláčková, Eva Landová, Daniel Frynta

**Affiliations:** 1Department of Zoology, Faculty of Science, Charles University in Prague, Viničná 7, 128 43 Prague, Czech Republic; stankohe@natur.cuni.cz (H.S.); PeleskovaSarka@seznam.cz (Š.P.); kristyna.sedlackova@email.cz (K.S.); 2National Institute of Mental Health, Topolová 748, 250 67 Klecany, Czech Republic; Markii47@seznam.cz (M.J.); frynta@centrum.cz (D.F.)

**Keywords:** negative emotions, fear, disgust, animals, human–animal relationship

## Abstract

**Simple Summary:**

This study focused on human negative emotions (fear and disgust) evoked by animals and aimed to find the most fear- and disgust-eliciting species/morphotypes across the animal kingdom. The examined stimuli included the top-scoring animals from previous experiments (two sets of 34 pictures). These were evaluated by Central European respondents representing WEIRD societies (Western, educated, industrialized, rich, and democratic). The respondents ranked the stimuli according to both elicited emotions. The results show that the most feared animals are mostly large vertebrates, such as carnivorans (bear, lion, tiger, etc.), ungulates (rhinoceros, hippopotamus, etc.), sharks, and crocodiles. Smaller fear-evoking vertebrates are represented by snakes, and invertebrates are represented by spiders and scorpions. The most disgust-evoking animals are human endo- and ectoparasites (e.g., tapeworm and tick) or animals visually resembling them (e.g., earthworm). A deeper understanding of negative emotions and the differences between fear and disgust elicited by animals might also help in nature conservation efforts, so that we know what factors or features may affect a negative attitude toward certain animals.

**Abstract:**

Animals have always played an important role in our everyday life. They are given more attention than inanimate objects, which have been adaptive during the evolution of mankind, with some animal species still presenting a real threat to us. In this study, we focused on the species usually evaluated as the scariest and most disgusting in the animal kingdom. We analyzed which characteristics (e.g., weight, potential threat for humans) influence their evaluation in a nonclinical Central European WEIRD population (Western, educated, industrialized, rich, and democratic). The tested animals were divided into two separated sets containing 34 standardized photos evoking predominantly one negative emotion, fear or disgust. The pictures were ranked according to their emotional intensity by 160 adult respondents with high inter-rater agreement. The most fear-eliciting species are mostly large vertebrates (e.g., carnivorans, ungulates, sharks, crocodiles), whereas smaller fear-evoking vertebrates are represented by snakes and invertebrates are represented by arachnids. The most disgust-evoking animals are human endo- and ectoparasites or animals visually resembling them. Humans emotionally react to fear-evoking animals that represent a real threat; however, identifying truly dangerous disgust-evoking animals might be harder. The results also support a somewhat special position of snakes and spiders.

## 1. Introduction

Throughout the evolution of humankind, animals have always played an important role in our everyday life, particularly as a source of food, clothing, or information; however, they have also represented a real threat to human health and lives. Regardless of our preferences for different species, animals attract human attention (this phenomenon is also known as biophilia [1]). In addition, many studies have shown that such increased attention to animals is also manifested at the neural level [2,3], as the primate visual system is adapted to detect animals prior to inanimate objects [4]. In the context of human evolution, it was important for our ancestors to pay attention especially to dangerous animals. For example, the detection and subsequent fear reaction to a potential predator had to be fast enough to be adaptive [5,6]. Studies concerning the prioritization of animal stimuli show that fear-relevant stimuli are detected faster than neutral ones, e.g., snakes and spiders versus flowers and mushrooms [7] or lions versus impalas, even on the basis of their low-level features (spatial frequency and luminance [8]).

Thus, this increased attention is accompanied not only by positive emotions but also by negative ones. In this study, we focused on two negative emotions, fear and disgust. Both of these emotions are functionally adaptive as they help us navigate in the environment, identify a risk early enough to stay alive, and react to it appropriately. Fear is triggered in the presence of a predator or another kind of imminent threat. The subsequent physiological reaction launches a cascade of processes (involving activation of the sympathetic nervous system, e.g., elevated heart rate) enabling a relevant behavioral reaction, i.e., fight, flight, or freezing [9,10]. On the other hand, disgust should protect us against transmission of infections and diseases through disease avoidance behavior [11].

Both fear and disgust also play an important role in anxiety disorders including animal phobias [10,12,13], affecting 3.3–5.7% of the worldwide population [14,15]. The most common phobias are elicited by spiders (2.7% of the population [16]) and snakes (2–3% [17,18]).

In the past decades, numerous studies aimed to identify animals evoking strong negative emotions or phobias across the animal kingdom and examine fear- and/or disgust-eliciting animal stimuli from different perspectives. Ware et al. [19] and Davey [20] studied common animal fears and both found a category of highly feared animals that, however, do not pose a real threat of causing serious injuries to humans (called fear-relevant animals, e.g., rat, bat, snake). Disgust sensitivity was significantly correlated with the fear rating of fear-relevant animals, suggesting an important role of the disgust emotion in animal fears. Disgust sensitivity or general disgust score were also highly correlated with disgust-relevant animals (also called invertebrates or repulsive, e.g., slug, maggot, cockroach [20,21,22]), but not with predatory animals (e.g., lion, crocodile, shark [19,21,22]). These studies, however, worked only with a rating of the list of animals on a Likert-like scale according to fear. Although the final categorization of animals was not always fully consistent, the importance of further examining the role of disgust clearly emerged.

Using general verbal stimuli (such as “frog” or “snake”) may not be optimal for this type of study given the large morphological variability in animal taxa. It was previously demonstrated that, e.g., not all snakes are alike; some species/morphotypes may evoke fear, while others trigger disgust [23]. Moreover, photographs may reliably substitute live snakes in terms of eliciting emotions in a self-reported evaluation [24]. There have been quite a number of studies aiming to determine the species evoking the examined emotions most strongly in a given taxonomic group, which used visual stimuli, e.g., amphibians (disgust [25]), reptiles (fear and disgust [26]), snakes (fear and disgust [23]), or invertebrates (fear and disgust [27]). In other studies, positive emotions (aesthetic preferences/beauty) were examined (snakes [28,29]; birds [30]; mammals [31,32]). The results of Frynta et al. [25] and Janovcová et al. [26] suggested that, in the case of amphibians and reptiles, beauty and disgust form opposite ends of one axis. Although, as far as we know, this was not tested in other groups, this suggests that studies concerning animal beauty might be of use also in disgust-related research. Therefore, animals that were evaluated as the least beautiful ones might be considered disgusting to some extent. Using more visual stimuli rather than one word might result in more detailed insights into human–animal relations. These studies are, however, limited as they usually cover only one taxonomic group.

Possidónio et al. [33] used 120 pictures from 12 biological categories throughout the whole animal kingdom, but they examined dimensions other than basic emotions (e.g., valence, cuteness, dangerousness). Polák et al. [34] investigated fear and disgust elicited by pictures of the 25 most common phobic animals. Results of a redundancy analysis showed two axes describing the evaluation of fear and disgust, one of which reflected a general negative attitude toward animals, while the second was associated with specific fear from snakes and/or spiders. Studied animals formed five distinct clusters: non-slimy invertebrates; snakes; mice, rats, and bats; human endo- and exoparasites; farm/pet animals. Although these animals belong to the most common phobic animals, only spiders, snakes, and parasites evoked strong fear and disgust in the nonphobic population. Thus, research of animals eliciting the most intense fear and disgust in a nonclinical population remains incomplete.

In the present study, we selected top-ranking animals according to elicited fear and disgust from different taxa across the animal kingdom based on previous studies, and we did not focus only on phobic animals. The tested animals were divided into two sets according to the emotion they predominantly evoke (fear or disgust). We are aware that the same animals can elicit both negative emotions [34], but we aimed to select species evoking one distinct negative emotion only. Moreover, our previous experiences showed that some people might have problems distinguishing these two emotions when rating pictures, thus motivating the separate sets.

We focused on four main aims:Find out what animals across the whole animal kingdom evoke the most intense fear or disgust in the nonclinical population, i.e., determine the scariest and most disgusting animals.Discuss which characteristics of the tested animals influence the evaluation (e.g., body size, venomousness, physical dangerousness).Analyze how respondents agree on the evaluation of pictures, i.e., whether they fear or are disgusted by the same animals.Explore which characteristics of respondents influence their evaluation of animals (e.g., gender, education, keeping pets, scores of questionnaires).

## 2. Materials and Methods

### 2.1. Selection and Preparation of Stimuli

Two separate sets of pictures of animals were used to test human negative emotions elicited by animals: set I for fear-evoking animals and set II for disgust-evoking ones. Selected animals covered both vertebrates and invertebrates across the animal kingdom and belonged to the top-rated animals (the most disgusting or the most fear-evoking) in their respective taxa. We aimed to employ specifically the species eliciting one strong discrete emotion to avoid interference in the evaluation and interpretation of the results. The stimuli were selected according to previous studies as follows: first, we selected top-rated species from studies using visual picture stimuli, from Polák et al. [34] (red panda, rat, louse, roundworm, tapeworm, moth, fly larva, and cockroach), Rádlová et al. [23] (rattlesnake), Frynta et al. [25] (frog and toad), Janovcová et al. [26] (crocodile, turtle, lizard, coral snake, and blind snake), Peléšková [35] (ostrich, shoebill, eagle, condor, and guineafowl), Possidónio et al. [33] (elephant, lion, bear, wolf, tiger, shark, piranha, scorpion, crab, beetle, butterfly, giant panda, bat, tick, earthworm, leech, and rabbit), and Landová et al. [31] (rhinoceros, hippopotamus, buffalo, hyena, mole rat, and porcupine). However, the research covering all animal taxa has not yet been completed (mainly in the case of extremely variable invertebrates). Thus, we added additional species on the basis of our zoological knowledge (e.g., real threat) and previous experience with similar research to cover the full morphological and phylogenetic diversity (e.g., barracuda, hornet, grub, centipede, horsefly, and lamprey). We also took into account evolutionary and cultural aspects. For these reasons, we preferred European or African species (longer co-evolution with the mankind) and well-known, culturally important charismatic animals, e.g., a wolf as a representative of canines (both culturally and evolutionarily important) or a tick as a non-spider arachnid (locally well-recognized). According to the abovementioned criteria, we selected 28 candidate species eliciting a given emotion for each set (fear and disgust; for the full list of species, see Appendix A). For the purpose of further analyses, we defined a potential threat represented by these animals to contemporary humans. Animals considered as dangerous and a risk for health were those commonly known as currently causing problems or deaths, whereas rare cases were omitted.

Moreover, we added four pictures as control stimuli not evoking the respective tested emotion, but either eliciting the second negative emotion (i.e., fear for the disgust set or vice versa) or not eliciting any negative emotion. For the fear set, we selected the red panda (*Ailurus fulgens*) and a butterfly (*Papilio machaon*). Both are perceived as neither fear- nor disgust-evoking. The mole rat (*Spalax ehrenbergi*) and blind snake (*Xenotyphlops grandidieri*) are not fear-evoking but disgusting. For the disgust set, we used rabbit (*Oryctolagus cuniculus*), butterfly (*Iphiclides podalirius*), polar bear (*Ursus maritimus*), and rattlesnake (*Crotalus cerastes*). The former two are neither fear- nor disgust-evoking, while the latter two are not disgusting, but fear-evoking.

Lastly, we added the giant panda (*Ailuropoda melanoleuca*) and tiger (*Panthera tigris*) into each set as perception controls. The panda does not evoke negative emotions, while the tiger evokes fear, but not disgust [31,32,33]. We further treated the tiger as an additional candidate fear-eliciting stimulus, for the purpose of analyses of the fear dataset. Thus, the fear set included five non-fear-eliciting species, while the disgust set contained six non-disgusting ones.

Thus, each set contained a total of 34 animal species. For each selected species, we found a representative photo of an adult individual on the Internet or among our own resources (for the source list, see Appendix A). Only high-quality photos depicting the animal in full body and natural coloring were chosen. We adjusted the photographs to a standardized form, i.e., the animals were placed on a white background, adjusted to a similar position and comparable body size, and then printed in a 10 × 15 cm format.

### 2.2. Testing Procedure

#### 2.2.1. Participants

In total, 160 adult participants (108 women and 52 men) were included in this study which was performed in the Czech Republic. The respondents were of a Central European origin, mostly Czech citizens. We recruited them mainly from staff and students of Charles University in Prague, a grammar school Jeronýmova in Liberec, and their relatives (aged 18–82, mean age 27.92, median 23). All subjects gave their written informed consent in accordance with the Declaration of Helsinki and then filled a personal questionnaire about their gender, age, education (level and field of study), and kept pets. The study was approved by the research ethics committee of Charles University, approval number 2018/17.

#### 2.2.2. Questionnaires

All participants filled out a battery of the following questionnaires (in the Czech language). To test a nonspecific/general level of fear from animals, we created the Animal Fear Questionnaire (AFQ) containing 13 groups of animals included in both experimental sets. Participants rated each item according to perceived fear on a seven-point Likert-like scale (1 = no fear, 7 = extreme fear; see Appendix A). We then used two psychological questionnaires focused on fear of two specific animals causing the most common animal phobias, Spider Questionnaire (SPQ [17], translated by Polák et al. [36]) and Snake Questionnaire (SNAQ [17], translated by Polák et al. [18]). The last questionnaire was the Disgust Scale-Revised (DS-R [37], modified by Olatunji et al. [38], translated by Polák et al. [39]) evaluating disgust propensity (see also Table 1 for the overview of the abbreviations used in the study). In order to test the effect of respondents with high fear of snakes and spiders, the high-fear participants were defined as those scoring above the upper quartile on the SNAQ/SPQ scales as used in previous studies by Polák et al. (upper quartiles were computed for the Czech population: SNAQ score ≥ 8 [18]; SPQ score ≥ 16 [36]).

#### 2.2.3. Stimulus Ranking According to Fear or Disgust

To test the emotional response to animals by human respondents, we adopted a widely used method of sorting picture stimuli according to a given emotion (e.g., [23,40,41]). This method is optimal for studies examining the variability in elicited emotions as it maximizes differences among species, compared to, for example, picture rating on a Likert-like scale.

Each respondent evaluated both sets (with a short break between them). All pictures from one set were placed on a table in a random order. The participants were asked to order all pictures according to the given emotion, to make a pack of the pictures, where the most fear-/disgust-evoking animal is on the top of the pack and the least fear-/disgust-evoking animal is at the bottom. Afterward, the respondents selected a picture of the animal determining the threshold where they did not feel fear/disgust anymore, thus dividing the pack of pictures into two categories (truly fear-/disgust-evoking and non-fear-/disgust-evoking). No time limit was set for performing both tasks, and the order of the tasks (first fear or disgust ranking) was counterbalanced.

### 2.3. Statistical Analysis

The raw ranks were square root arcsine-transformed to increase the importance of the distribution tails and to improve normality. Mean values of the transformed ranks were further used as a scale for perceived fear and disgust of selected animals (the higher the value, the lower the perceived fear or disgust). The agreement in species ranking among the respondents was quantified using Kendall’s coefficient of concordance. Spearman’s coefficient was computed to examine the relationship between different parameters of fear and disgust evaluation. The significance of differences in mean ranks among individual species for each emotion was tested using the post hoc Friedman–Neményi test as implemented in the R package PMCMR [42]. A redundancy analysis (RDA; implemented in the R package vegan [43]) was performed to examine the contribution of respondent’s characteristics (gender, education, pets, negative experience with animals, and SNAQ, SPQ, DS-R, and AFQ scores) to the fear and disgust evaluation. The RDA is a multivariate analysis, which can detect a more complex relationship between a set of responses and explanatory variables. Statistical significance was confirmed by permutation tests. A Mann–Whitney U test was then used to compare the evaluation between the respondents with high and low fear of snakes and spiders. Lastly, linear models (LM) were used to test the effect of species’ characteristics (dangerousness, venomousness, health risk, and size) on their evaluation. Calculations were performed in R [44] and Statistica [45].

## 3. Results

### 3.1. Emotional Salience of the Examined Stimuli

The respondents ranked animals depicted on photographs according to the respective negative emotion (34 stimuli in each of two sets: fear and disgust). They also determined a threshold rank (range 1 to 30 and 1 to 32, mean number 16.56 and 15.64 for fear and disgust stimuli, respectively) separating fear-/disgust-evoking stimuli from those not eliciting the emotion. Thus, only around a half of the stimuli were evaluated as truly fear-/disgust-evoking by the majority of respondents. Mean animal ranks and percentage of respondents, who assessed each animal above the threshold, were highly intercorrelated (Spearman’s correlations: for fear, *R* = 0.991; for disgust, *R* = 0.985; both *p* < 0.0001; Figure 1; for more descriptive statistics of the stimuli, see Appendix A).

The top fear-eliciting stimuli were the shark, crocodile, snakes (rattlesnake, coral snake), large mammals (carnivorans and ungulates), and arachnids (scorpion, black widow, solifuge). In contrast, the most disgust-eliciting belonged almost exclusively to small invertebrates (mostly exo- and endoparasites or arthropods, such as the spider or centipede), with the exception of the Mexican mole lizard (*Bipes biporus*).

The tested stimuli significantly differed in their ranks (Friedman test for both emotions *p* < 0.0001; 64% and 66% of post hoc Neményi comparisons among candidate species were significant at *p* = 0.05, for fear and disgust, respectively; for more information see Appendix A). Mean ranks of non-fear-/disgust-evoking animals were significantly higher (i.e., lower level of fear/disgust) than those of the most fear-/disgust-evoking species. In the case of fear, 126 out of 143 such comparisons were significant (*p* < 0.05) and only two of the pre-defined fear-evoking animals, the shoebill (*Balaeniceps rex*) and European stag beetle (*Lucanus cervus*) ranked lower than one (for shoebill) and two (for stag beetle) non-fear-evoking animals. For disgust, 154 out of 168 comparisons were significant. Only the rattlesnake (*Crotalus cerastes*), predefined as a non-disgust-evoking but fear-evoking stimulus, ranked higher than seven disgust-evoking animals and, thus, elicited some level of disgust.

Next, we performed linear models to analyze the effect of selected characteristics of examined animal species on their fear and disgust evaluation by the respondents. Dangerousness to humans, venomousness, and the weight category were introduced as explanatory variables to explain mean fear ranks (arcsine-transformed). The final reduced model (reduced according to the Akaike’s information criterion (AIC)) revealed only a significant effect of dangerousness (*p* = 0.0004); the model explained 39% of the full variability (*R*^2^ = 0.3909). For the disgust ranking, weight category and health risk (risk of disease or infection transfer) were used as explanatory variables. The final reduced model explained 27% of variability; however, the effect of the weight category was only marginally significant (*p* = 0.0511), while the real health risk to humans remained insignificant.

We also examined which animals were chosen in the first three places during the ranking (see Appendix A). This closely corresponded to the mean ranks with exception of the spider; in both sets, the spider took the second place when analyzing which animal was the most frequently chosen as the most fear-/disgust-evoking (first picture during the ranking). However, it was evaluated as less disgusting according to its mean rank (seventh place according to the fear ranking, 10th according to disgust), suggesting a special human relationship with spiders.

### 3.2. Agreement among Respondents

The respondents had no trouble ranking both sets of pictures and strongly agreed on the final order. Kendall’s coefficient of concordance was W = 0.526 for fear ranking and W = 0.697 for disgust (both *p* < 0.0001). The agreement among respondents between high- and low-fear sub-groups was comparable (Kendall’s W set fear: high SNAQ W = 0.51, low SNAQ W = 0.553, high SPQ W = 0.542, low SPQ W = 0.534; set disgust: high SNAQ W = 0.679, low SNAQ W = 0.719, high SPQ W = 0.77, low SPQ W = 0.689; all *p* < 0.0001). This allowed us to further compare these sub-groups (see Mann–Whitney U test below).

### 3.3. Variability among Respondents

We further performed a redundancy analysis (RDA) to examine the contribution of respondents’ characteristics to fear and disgust evaluation (raw data). We included the following explanatory variables for both emotions: gender, level and field of education (biological or not), keeping a pet, having any negative experience with animals, and whether this negative experience has affected the respondent’s life. Moreover, we added total scores of the SNAQ and SPQ questionnaires, subscale scores of the DS-R (core, animal reminder, and contamination disgust [38]), and scores of individual questions of the AFQ, as each of the questions deals with different animal taxa/group (see Table 1 for an explanation of the abbreviations). The RDA for fear generated seven constrained axes that explained 16.31% of the full variability in animal ranking according to fear. Sequential “Type I” ANOVA (*n* permutations = 20,000) revealed that the effects of the total SNAQ score and individual questions AFQ 4 (fear of spiders), AFQ 7 (stinging insects), and AFQ 8 (large carnivorans) were significant at the *p* < 0.0001 level. AFQ 10 (birds of prey) was significant at the *p* < 0.001 level, and AFQ 11 (sharks) and the level of education were significant at the *p* < 0.01 level (for the visualization, see Figure 2). In case of disgust, the RDA extracted four constrained axes explaining 10.98% of variability in animal ranking according to perceived disgust. ANOVA revealed that the effects of the total SNAQ score and questions AFQ 2 (mice and rats) and AFQ 4 (spiders) were significant at the *p* < 0.0001 level, and the level of education was significant at the *p* < 0.001 level (Figure 2).

Because both rankings were significantly affected by respondents’ fear of snakes (SNAQ) and spiders (AFQ 4), we used Mann–Whitney U tests (Bonferroni-corrected) comparing the raw mean ranks of each species in respondents with high and low fear of snakes and spiders to identify the species that substantially contributed to the differences between these groups of respondents. In the fear set, only the aquatic coral snake (*Micrurus surinamensis*) and Madagascar blind snake (*Xenotyphlops grandidieri*) were ranked as more fear-evoking by respondents with high fear of snakes. The Iberian solifuge (*Gluvia dorsalis*) and southern black widow (*Latrodectus mactans*) were ranked as more fear-evoking and the hippopotamus (*Hippopotamus amphibius*) as less fear-evoking by respondents with a high fear of spiders. In the disgust set, the bed bug (*Cimex leuctularius*) was ranked as less disgusting and the sidewinder rattlesnake (*Crotalus cerastes*) as more disgusting by respondents with a high fear of snakes. Lastly, the trapdoor spider (*Aptostichus sp.*) was reported as more disgusting by respondents with a high fear of spiders (all *p* < 0.0015, i.e., the *p*-level defined by the Bonferroni correction). When the species whose evaluation differed between high- and low-fear respondents were excluded, the overall order remained almost unchanged. Moreover, after excluding these species from the datasets, the agreement among the respondents was only a little higher (fear W = 0.559, disgust W = 0.726, both *p* < 0.0001). As it was already quite high in the original dataset, we decided not to exclude these species from the subsequent analyses.

Lastly, we analyzed correlations between scores of general questionnaires (i.e., DS-R and AFQ) and a number of chosen animals within the threshold, as it may also reflect a level of fear and disgust elicited by the tested animals. The results were significant with more sensitive people choosing slightly more animals, but rather low (Spearman’s correlations for fear: DS-R *R* = 0.1724, *p* = 0.0293; AFQ *R* = 0.3621, *p* < 0.0001; disgust: DS-R *R* = 0.3044, *p* = 0.0001; AFQ *R* = 0.2411, *p* = 0.0021).

Thus, in summary, the main differences were in the ranking of snakes and spiders by the highly fearful participants, and these analyses supported our conclusion that the variability among the respondents was overall quite low.

## 4. Discussion

This study presents a collection of the most fear- and disgust-evoking species across the animal kingdom. The top-rated animals according to fear belonged mostly to large vertebrates, such as large carnivores, ungulates, sharks, and crocodiles. Smaller fear-evoking vertebrates were represented by snakes, and invertebrates were represented by arachnids. The most disgust-evoking animals were human endo- and ectoparasites or animals visually resembling them.

In general, our results suggest that fear-evoking animals are predominantly large, with visible weapons, or venomous. Large animals usually pose a real threat to humans (although the number of actual encounters and fatalities might differ). For venomous species, the body size is not determinative; thus, even considerably small animals might evoke fear. However, venomousness within a single morphotype (visually similar species) might be far more different for individual species, considering moreover the existence of mimicry. Thus, animals with no toxicity but resembling the venomous ones may be feared as well, although this spontaneous cognitive categorization is not yet fully understood. It is also worth noting that large and venomous animals do not form clear distinct categories according to their fear ratings; in other words, large animals do not evoke greater fear than venomous ones and vice versa.

For disgust, the pattern within the most disgust-eliciting animals is not as obvious. This emotion is predominantly evoked by small invertebrates; however, the effect of actual threat represented by the tested animals was not significant, which is supported by previous studies on the role of generalization and transfer of the contamination or offensiveness [46].

Lastly, our results showed that the human attitude toward snakes and spiders (i.e., animals causing the most prevalent zoophobias) is somewhat special, because they might evoke both emotions to some extent. Thus, they are discussed separately.

### 4.1. Fear

On the basis of the results, we propose three possible explanations why a particular animal species is feared by humans:Evolutionary experience with an important predator of human/human ancestors and, thus, possible co-evolution.An animal representing an actual danger for contemporary humans who have direct or indirect experience with it. However, this category could also include cultural influences (e.g., movies, legends, and mythology), which are not always based on the real threat presented by the animal.Presence of features that might be perceived as weapons, features of possible aggressive motivation, or fighting ability (e.g., teeth, claws, large body size, or aposematic coloration).

Thus, an animal is evaluated as fear-evoking if it fits at least one of the categories above. Most of the feared animals, however, meet more than one condition. The relevance of these criteria for the top-ranked animals is further discussed in the order according to their fear rating.

#### 4.1.1. Sharks (Lamniformes)

The shark was evaluated as the most-fear evoking of all the animals included in the present study. However, neither human nor nonhuman primates have ever represented an important food source for sharks [47]. Despite a considerable potential of most shark species to seriously harm or kill humans, which is further supported by sensationalist negative portrayal in the media and culture [48], the annual global average is four fatalities [49]. Thus, the most important factor in the evaluation seems to be the sharks’ appearance—they possess very strong features, such as a large body size or sharp teeth visible, even without a threat. Moreover, sharks are quite evolutionarily distant from humans and live in a different type of habitat, i.e., water. Such unfamiliar environment and different ecology (unpredictable behavior) may, thus, cause delayed detection, greater anticipation anxiety, and fear of the unknown. This is consistent with previous studies showing that human attitudes are influenced by the level of biological and behavioral similarity with a given species [50].

#### 4.1.2. Crocodiles (Crocodylia)

During the evolutionary history, crocodiles and human ancestors inhabited the same regions and, although humans were probably not their primary prey, fossil remains suggest that at least some human ancestors were hunted by crocodiles [6]. The evolutionary relevance of fear of crocodiles is also supported by the study of Cook and Mineka [51], where a crocodile toy was successfully used as a fear-relevant stimulus for naïve rhesus monkeys. Crocodiles also figured in myths of many cultures and are of a totemic significance to this day [52]. Currently, crocodiles kill an estimated 1000 people every year [53]. Similarly to sharks, crocodiles possess fear-evoking visual features and live in a semiaquatic habitat, where it is often hard to navigate for humans.

#### 4.1.3. Snakes (Squamata)

Snakes not only represent important predators of primates [5], but also were probably one of the first predators of placental mammals [54,55], which suggest long co-evolution and strong predispositions for fear of snakes in human minds. Today, venomous snake bites cause between 81,410 and 137,880 deaths annually [56], and ophidiophobia is one of the most prevalent zoophobias in humans [17,18]. Given the fact that many snakes can be deadly, it would probably be advantageous to generalize and fear all the snakes the same rather than taking a risk of being bitten by a venomous snake. However, recent studies have shown that not all snakes are perceived as one category in terms of experienced emotions (not only fear, but also disgust; see below [23,57]. Some snake morphotypes are feared more than others. For example, Polák et al. [34] found that a viperid snake was evaluated as more fear-evoking than a nonvenomous colubrid snake. It also scored higher than a venomous coral snake with aposematic coloration in the present study, which further supports the hypothesis that not all snakes are perceived alike (for more information on this topic, see also [23,29]).

#### 4.1.4. Big Cats and Bears (Carnivora)

Members of the order Carnivora were important predators of primates and human ancestors in Africa (especially the suborder Feliformia and, less frequently, Caniformia [47,58]). Even to this day, fatal encounters do occur, e.g., at least 563 humans were killed between January 1990 and September 2004 by lions in Tanzania [59], and 398 people were killed by bears (excluding the polar bear) in the 20th century [60]. Although tigers were not co-evolutionary human predators, the level of fear elicited by tigers in this study was comparable to lions; bears and tigers certainly kill a significant number of people today—12,599 reported kills in the 20th century [60]. Interestingly, big cats and bears also belong to the most beautiful animals as evaluated by human respondents [31,32]. This might seem contradictory, but it was demonstrated that dangerous species capture and maintain human attention more quickly than harmless animals [8], which may also affect the perception of animal beauty. Moreover, cats and bears possess features such as a large head, big eyes, and round ears that resemble juvenile traits (baby schema [61]) and are perceived as cute and beautiful [31]. In particular, in big cats, the positive attitude often predominates (even in areas where wild cats live [62]; compare with canids below). This might eventually lead to underestimating the real threat they pose, which can be demonstrated by the large number of big cats being kept as exotic pets and a non-negligible number of attacks by them [63].

#### 4.1.5. Arachnids (Araneae, Scorpiones, Solifugae)

Although the most dangerous arachnids (both today and evolutionarily) are scorpions with 3250 fatalities every year [64], spiders compared with scorpions were evaluated as more fear-eliciting and more frequently trigger a phobic reaction. Arachnophobia is one of the most prevalent animal phobias [16]. Although the term could indicate otherwise, it is almost always restricted to spiders, and fear of any other well-known arachnids such as scorpions is surprisingly nearly absent in the literature (however, see [65], where scorpion fear scores were equal to or significantly higher than spider fear scores). This topic would apparently require more research and will be addressed in our next study. While spiders were represented by a deadly venomous species (southern black widow) in this study, many other spider species inhabiting the area of human evolutionary origins are not dangerous for humans. Spider venom may cause infections and tissue damage [66]; however, globally, spiders pose a minimal threat to human life, as spider bites kill less than 0.001 per million people [67]. They could, however, represent a significant threat locally; for example, in the United States of America (USA), spiders kill six people annually [68]. Thus, fear of spiders could be considered as at least partially reasonable, but co-evolution with primates or humans is rather unlikely. The negative attitude toward spiders is also influenced by culture and the emotion of disgust, especially in Europe and other Western countries [69] (for more discussion on this topic, see below).

#### 4.1.6. Hyenas and Wolves (Carnivora)

Primates and human ancestors (especially their juveniles) in Africa were hunted by hyenas and wild dogs. As hominids dispersed, they encountered larger wolfish canids representing truly formidable predators [58]. Although fatalities may still occur today (including deaths caused by rabies transmitted by wolves [60]), well-documented cases are globally rather rare and such high fear is caused more by folklore than a real threat [70]. However, human attitude to canids is surprisingly ambivalent. Treves and Bonacic [71] proposed a dual-response hypothesis of attraction or aversion that are expressed independently toward domestic and wild canids (i.e., positive attitude toward domestic dogs as companions and helpers and negative attitude toward wild canids as competitors, predators, or disease vectors).

#### 4.1.7. Large Herbivores (Artiodactyla, Perissodactyla, Proboscidea)

Throughout evolutionary history, human ancestors and large herbivores have long co-existed in Africa. Although herbivores do not feed on humans, there is an evidence of increasing herbivore–human conflicts over land and other resources with a growing human population [72]. Between 1923 and 1994 in Uganda, 33.5% out of 636 reported causalities were caused by large herbivores (especially elephants, hippopotami, and buffaloes [73]). The fatality rate from 49.2% (buffaloes) to 86.7% (hippopotami; this is even more than lions with 75%) proves that their large body size (with their weight over 1000 kg, they are also called megaherbivores) and weapons represent a real threat to humans. European respondents also evaluated these large animals as fear-evoking, even if they do not encounter them in the local nature. However, the first impression of many respondents was mostly positive, and it took a second thought to consider these animals as dangerous while evaluating picture stimuli. People with higher biological knowledge tend to rate them as more fear-evoking (especially elephant and rhinoceroses), which might suggest a cultural transfer of learned behavior.

#### 4.1.8. Piranha (Characiformes)

Rather strong fear elicited by a South American fish, the piranha, is also noteworthy, as co-evolution with humans is unlikely and it does not easily cause fatal injuries to humans (although tissue loss and extensive bleeding occur [74]). Despite the folklore and movies, piranhas rarely attack large animals including humans (whether individually or in piranha schools [75]). Piranhas are also rather small compared to, for example, highly scoring sharks, and, although they have sharp teeth, these are not visible while swimming. Given the lower fear score of two other large carnivore fish included in the study (the great barracuda (*Sphyraena barracuda*) and goliath tigerfish (*Hydrocynus goliath*)), folklore might play a predominant role in fear of piranhas.

#### 4.1.9. Insect (Hymenoptera)

Lastly, the evaluation of hymenopteran insects should be evaluated, as bees, wasps, and hornets belong to the most frequent killers among venomous animals (USA [68]; Czech Republic [76]). Both wasps and bees were ranked within the top 10 of animal fears in Davey [20] and were also included in Polák et al. [34]. However, the hornet ranked 21st place in the current study. This might be caused by the fatality rate in comparison with other examined animal stimuli. In this case, the danger is rather affected by the incidence of stings; moreover, serious complications occur especially in hypersensitive or allergic people [77].

### 4.2. Disgust

Both fear and disgust play an important adaptive role in dealing with fundamental life tasks [78], yet they differ in their mechanisms. Disgust might be interpreted as fear of contamination and disease transmission [11]; however, as results of this study suggest, identifying an animal presenting a real disease- or contamination-associated threat can be hard. In general, disgust is evoked especially by invertebrates (see also [79]) that are highly variable in terms of the number of species and morphology, which complicates the identification of the risk level. The link between disgust-eliciting features and real danger might be questionable or even false. Thus, disgust might be evoked by animals that are not dangerous to humans, but resemble other disgust-eliciting stimuli, because they share similar features. On the other hand, there are also dangerous animals not eliciting any or low disgust. Such an example might be mosquitoes, as the diseases they spread make them the deadliest animals in the world [80]. Mosquitoes, however, do not evoke any disgust [81]; thus, they were not even included in this study. 

In case of disgust, the assessment of evolutionary importance might be difficult. Due to all the reasons mentioned above, the explanatory categories used for the fear evaluation are not applicable for disgust. On the basis of the results, we hypothesize three theoretical explanations for how an animal might acquire its disgust-evoking status, the first two of which are in agreement with Davey [82]:Animals representing a real threat by spreading diseases or infections.Animals possessing visual features resembling primary disgust-evoking stimuli (such as animals perceived as slimy) but representing no or very little danger to humans.Venomous animals that arguably evoke both emotions, fear and disgust.

Unlike fear-eliciting animals, the criteria rarely overlap for disgust stimuli. The relevance of these criteria for the top-scoring animals is further discussed according to their disgust rating.

#### 4.2.1. Endoparasites (Cyclophyllidea, Ascaridida)

The tapeworm (*Taenia solium*) is responsible for the highest number of deaths of all human endoparasites (cysticercosis, 26,000 fatalities in 2016 [83]) and was also evaluated as the most disgust-eliciting animal both in this paper and the previous study of Polák et al. [34]. The roundworm also scored high in both studies, which poses a serious threat as a disease vector (i.e., ascariasis, 6000 deaths in 2016 [83]). Intestinal worms represent a huge medical problem with approximately 1.5 billion people infected worldwide (almost one-quarter of the entire population), most of them from the poorest countries [83]. This supports the role of parasites in the disgust-motivated behavioral immune system and as a strong selective pressure in human evolution [11,84].

#### 4.2.2. Ectoparasitic Insects and Arachnids (Ixodida, Phthiraptera, Siphonaptera, Hemiptera)

The next highly scoring animals were hematophagous ectoparasites, most of which transfer life-threating diseases: tick (e.g., hemorrhagic fever, Lyme disease, or tick-borne encephalitis), louse (typhus or relapsing fever), and flea (plague [85]). It is noteworthy that a tick was evaluated as the most disgust-evoking ectoparasite, which might be affected by several factors; a picture of fully sucked tick was used (its shiny pale engorged body might be a visual disgust-eliciting stimulus per se, or the respondents are aware that it is full of human blood). Ticks are very common in Central Europe and there is an ongoing campaign to raise awareness about their dangerousness, as the Czech Republic has the highest incidence of tick-borne encephalitis in Europe [86]. Only bedbugs have not yet been demonstrated to be an infectious disease vectors, although there are candidate pathogens potentially transmitted by them [87]. Nevertheless, a bedbug bite can be quite painful, whereas anaphylaxis or severe anemia have also been reported [88]. Moreover, bedbugs are seen as indicators of filth and poor hygiene [89]. This is not to mention the blood-sucking, which alone might be perceived as uncanny, dangerous (fear of losing too much blood), or disgusting.

#### 4.2.3. Leeches (Arhynchobdellida)

The role of leeches as vectors in human diseases has been discussed several times but has not yet been proven [90]. However, complications from secondary infections to severe anemia [91] and even death may occur if a leech enters the respiratory tract [92]. Moreover, leeches are slimy and worm-like, and most of them are blood-sucking parasites, which may contribute substantially to their disgusting status.

#### 4.2.4. Larvae (Coleoptera, Diptera)

Although there is a high variability in larval biology and potential threat, most of them share similarities in their morphology—often pale worm-like body with reduced eyes. Larvae either might resemble intestinal worms [81] or are associated with rotten meat and corpses [93]. The resulting generalization is probably the reason why all three larvae included in the present study were evaluated as highly disgusting regardless of the disgust relevance (with a harmless grub scoring the highest).

#### 4.2.5. Centipedes (Scolopendromorpha)

The case of centipedes is somehow similar to spiders; they possess venomous glands, while larger species (e.g., Scolopendromorpha that was also included in the current study) are able to bite through human skin and cause severe pain and swelling, rarely anaphylaxis, or even death [94,95]. Although the effect is usually not life-threatening, one might argue that fear would be an appropriate emotion. Centipedes do evoke some fear, but disgust predominates (own unpublished data). This may be caused by its long, almost worm-like shiny body, unusually high number of limbs, and uncanny, unpredictable movements, as reported by the respondents. Unlike spiders, centipede phobia is not usually reported.

#### 4.2.6. Mexican Mole Lizard and Earthworm (Squamata, Opisthopora)

A Mexican mole lizard and earthworm fit undoubtedly the second explanation category (together with blind snakes (Typhlopidae) and other underground animals not included in the present study, such as caecilians [25]). They are harmless to humans but evoke disgust because of their slimy, worm-like appearance. However, they scored slightly lower than intestinal worms and larvae. Prokop et al. [84] showed that disgust of disease-irrelevant insect larvae/earthworm correlated with parasite avoidance score in the same manner as disease-relevant parasites. Moreover, a Mexican mole lizard with such an unusual number of legs (two) might be perceived as injured or disabled.

#### 4.2.7. Spiders (Araneae)

Spiders seemed to trigger intensive emotions in both experimental sets and, thus, evoked both fear (discussed above) and disgust in the respondents. Similar results were obtained in previous studies in nonclinical [34,96] and phobic participants [97]. Spiders were also associated with a disease spread (e.g., plague) during the Middle Ages in Europe [69]. However, the respondents usually stated that its numerous legs, movements, unpredictability, or unnaturalness make a spider disgusting, as opposed to being dirty or infectious [98,99]. Further research is needed to answer the question whether there are morphotypes of spiders evoking predominantly one emotion or rather a combination of both emotions of fear and disgust.

#### 4.2.8. Frogs (Anura)

Although some frogs may be dangerous even for humans, they do not evoke fear [100,101]. Many frog species of the family Dendrobatidae produce some of the most toxic alkaloid poisons ever known [102]. However, they have also contrasting aposematic coloration that was evaluated as beautiful by human respondents, not evoking any disgust [25]. Thus, it rather seems that different frog morphotypes are perceived as disgusting because of their appearance, which includes a round slimy body, often with warts, small eyes, and dull coloration [25].

#### 4.2.9. Snakes (Squamata)

Previous studies suggested that there are fear- and disgust-eliciting snakes [103], which was partially confirmed in this study. However, the predominant emotion was not fully discrete, especially in the disgust evaluation; although no snake scored in the top animals, both snakes evoked at least some level of disgust (even the sidewinder rattlesnake, which was for the purpose of the study described as primarily fear-evoking and non-disgust-evoking). This suggests that human perception of snakes might be negative in general regardless of the tested emotion, especially in high-fear respondents (who may not be able to distinguish between the emotions [103]). The context of other experimental stimuli is also important, as particular snake species (i.e., rattlesnake) might be relatively less disgusting than others; however, when presented in the context of other animals, some level of disgust might be evoked. In terms of features affecting their disgusting status, the respondents’ persisting belief that snakes are slimy is interesting, despite already knowing that this belief is false (even when they touched a snake themselves; own unpublished data).

### 4.3. Agreement among Respondents

The results revealed a significant congruence among the respondents in both experimental sets. The agreement was quite high compared to our previous studies using the methodology of picture ranking [24,41,104,105]. Such high agreement can be found when evaluating emotionally salient and evolutionarily relevant stimuli such as snakes, where human respondents strongly agree on which species (or morphotypes) are fear-evoking and/or beautiful, even cross-culturally [29,40]. The congruence might be compared also with studies testing perceived beauty of animals, as previous research showed that beauty and disgust are strongly negatively correlated and might even represent the opposite ends of the same axis, at least in some animal taxa (amphibians [25]). Moreover, the agreement was higher when evaluating disgust than fear. This might be due to the more complicated recognition of the stimuli and their actual risk to human health compared to widely known charismatic megafauna and other fear-evoking animals. Such stimuli might then be evaluated predominantly on the basis of their visual appearance, just like the evaluation of beauty.

### 4.4. Threshold

A new experimental method was established in the present study. After the stimulus ranking, the respondents selected a picture of the animal determining the threshold where they did not feel fear or disgust anymore, thus separating fear-/disgust-evoking stimuli from those not eliciting the emotion. This method highly correlated with the ranking (mean animal ranks with the percentage of respondents who assessed each animal above the threshold) and, moreover, supported the stimulus selection, as each experimental animal elicited the given emotion at least in some respondents. Although the respondents sensitive to animal fears or disgust tended to choose slightly more animals in this task, the effect was rather low, and the tested stimuli were also emotionally salient for normal or the low-fear/disgust population.

### 4.5. Effect of Respondents’ Characteristics

We found a significant effect of fear of snakes (as measured by the SNAQ questionnaire), fear of spiders (AFQ 4), and level of education. Heightened fear of snakes or spiders quite commonly occurs in the nonclinical population [34]. People with higher levels of such fear evaluated the respective animals as both more fear- and disgust-evoking (often scoring the first place); however, the remainder of the ranking remained comparable to the respondents with average/low fear. The effect of education was rather small with only three animals scored higher by the respondents with higher education (rhinoceros and elephant for fear; tick for disgust; all relevant stimuli in their emotional category). Interestingly, the overall level of education was more important than biological education, which remained insignificant. This might suggest that higher education of any field correlates with higher general knowledge, including a potential threat represented by animals. However, the study was not designed to examine the effect of education in the first place; thus, we do not want to overgeneralize these results.

The effect of gender was not significant for any of the tested emotions. The research shows that women are, in general, more sensitive to animal fears [34,37]; however, gender differences were usually negligible in studies using the methodology of ranking and comparing the animals rather than drawing comparisons between the respondents (i.e., relative order vs. absolute level of fear [29]; for a review on gender differences in human-animal studies, see [106].

### 4.6. Limitations of the Study

All the participants belong to the WEIRD society (Western, educated, industrialized, rich, and democratic) and are influenced by this culture and modern media. They represent a typical Central European population. Although we do not wish to overgeneralize our results, we believe that the sample was adequate for the purpose of this study. This is in concordance with previous studies using similar methodology which showed that the effects of the respondents’ sociodemographic characteristics on the emotional evaluation of animals in Western countries are usually rather small (e.g., [25,32,104,106]). A cross-cultural comparison was not an aim of the study, but it will be conducted in future research. The number of respondents was also influenced by the method of data collection. First, we decided to perform an experiment in contact with individual respondents, which allows more clarifications or corrections, and the results are considered to be more reliable. However, it complicates achieving a larger sample size compared to, for example, online surveys. Second, the rank-ordering method is optimal for evaluating relative differences in mean ratings of animal stimuli, but it is even more time-consuming.

## 5. Conclusions

This study included top-scoring animals from the previous experiments that aimed to find the most fear- and disgust-eliciting animals from a single taxon or group. Although the selected representatives were strongly negatively evaluated in their respective groups, the variability in perceived levels of fear and disgust in the final experimental sets was still rather high. i.e., the most fear- or disgust- eliciting species among birds might not be a strong stimulus compared to large predators or parasites. This variability among presented stimuli allowed the respondents to rank the animals without major problems and with a significant congruence.

We were, thus, able to find the most fear- and disgust-eliciting animals across the animal kingdom. Moreover, we selected the stimuli evoking the respective emotion in most of the respondents to be used in subsequent experiments. We aim to further examine these emotions evoked by animals using somewhat more objective methods, such as measuring physiological responses and eye-tracking, and to compare the self-reported evaluation of the stimuli in children.

Deeper understanding of negative emotions and differences between fear and disgust elicited by animals might also help in the nature conservation effort, so that we know what factors or features may cause a negative attitude toward certain animals. A discrepancy in evaluating potential threat of fear- and disgust-evoking animals also emerged. These results open new possibilities of future research in human ability to create cognitive categories and generalization according to the visual similarity.

## Figures and Tables

**Figure 1 animals-11-00747-f001:**
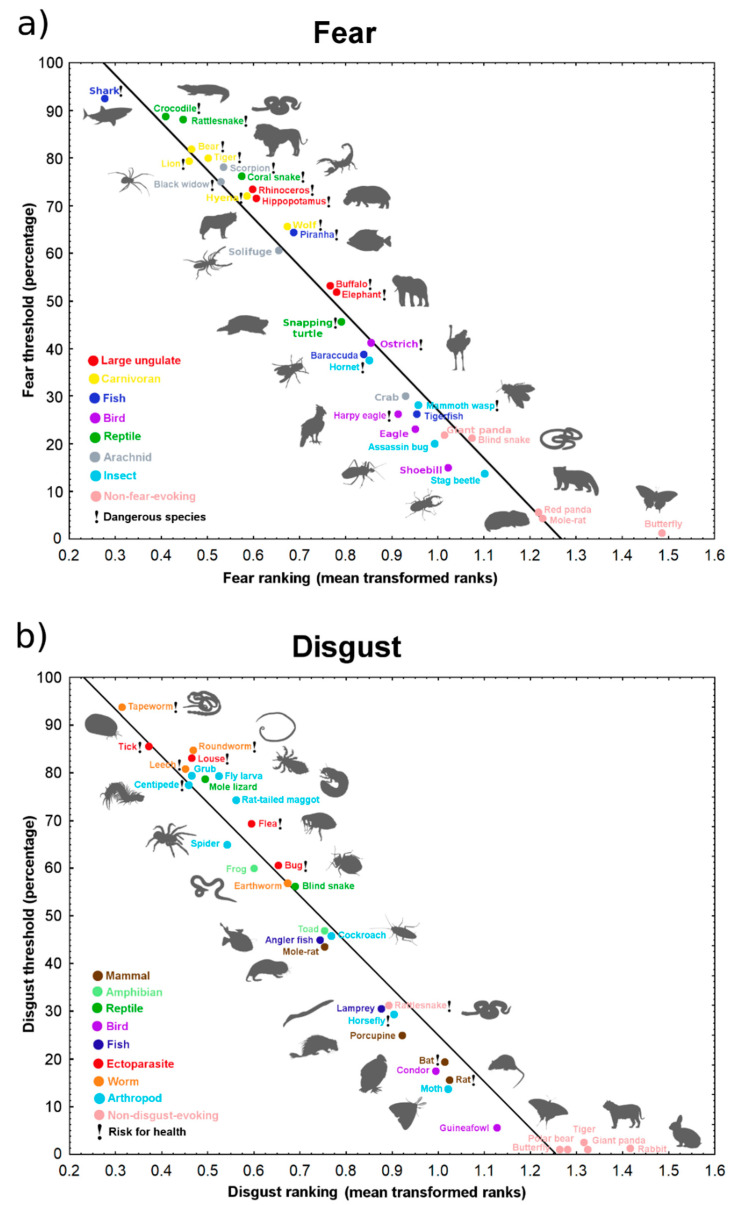
The correlation of animal ranking (mean square root arcsine-transformed ranks) and percentage of respondents, who evaluated each animal as truly (**a**) fear-evoking (Spearman’s *R* = 0.991, *p* < 0.0001) and (**b**) disgust-evoking (Spearman’s *R* = 0.985, *p* < 0.0001).

**Figure 2 animals-11-00747-f002:**
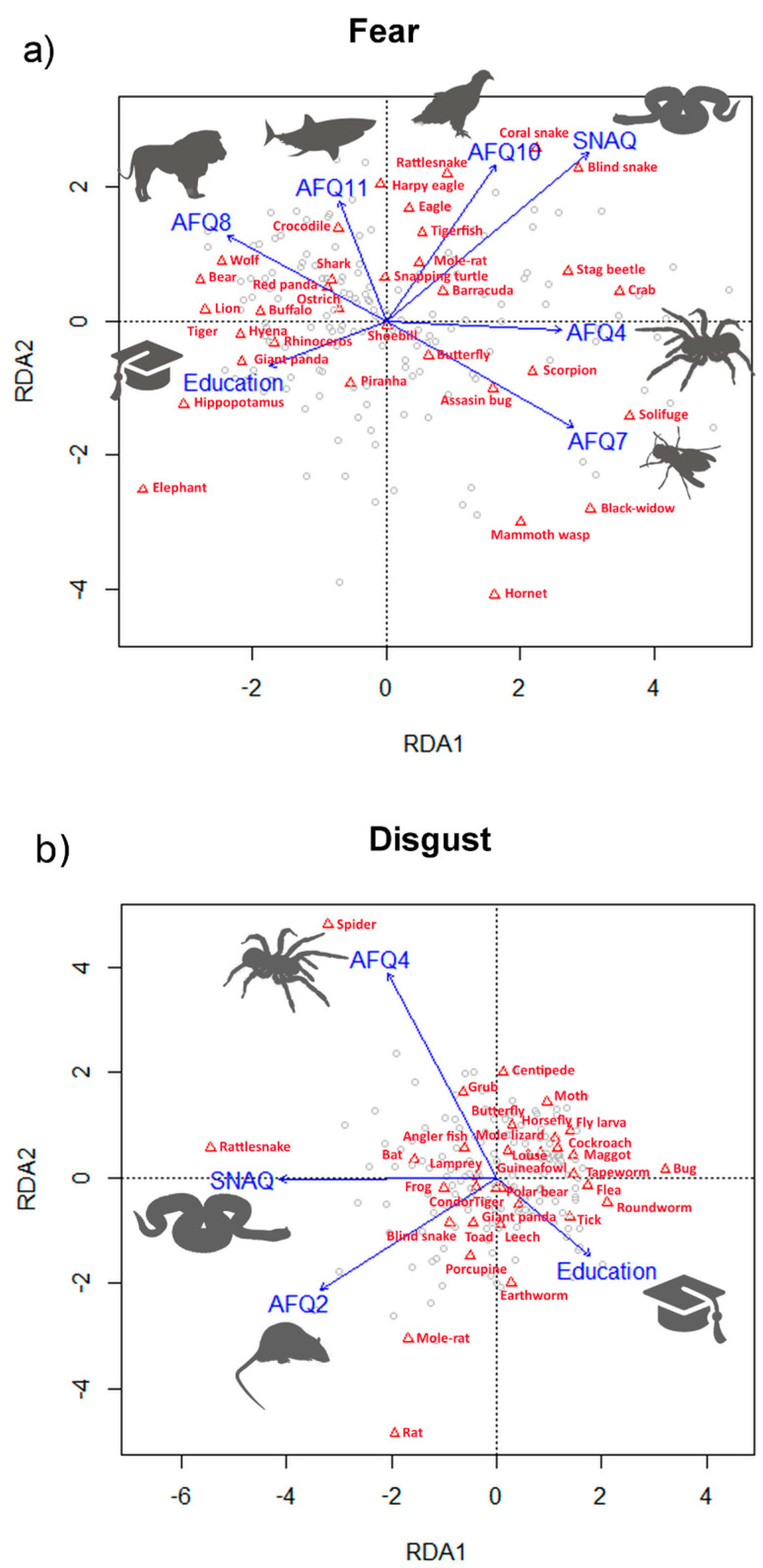
Redundancy analysis (RDA): visualization of the effect of respondents’ characteristics and questionnaire scores on (**a**) fear and (**b**) disgust rankings of the tested animals (red triangles). (**a**) The analysis explained 16.31% of the full variability. The main predictors are visualized using respective silhouettes (significant AFQ questions: large carnivorans, sharks, birds of prey, spiders, and stinging insects; fear of snakes (SNAQ); level of education). (**b**) The analysis explained 10.98% of the full variability. The main predictors are visualized using respective silhouettes (significant AFQ questions: mice and rats, and spiders; fear of snakes (SNAQ); level of education).

**Table 1 animals-11-00747-t001:** Overview of the abbreviations used in the study.

Abbreviation	Variable	Definition
SNAQ	Snake Questionnaire	Questionnaire measuring fear of snakes
SPQ	Spider Questionnaire	Questionnaire measuring fear of spiders
DS-R	Disgust Scale-Revised	Questionnaire evaluating disgust propensity
AFQ	Animal Fear Questionnaire	Questionnaire testing a nonspecific/general level of fear from different animal groups
W	Kendall’s coefficient of concordance	Nonparametric statistic used for assessing agreement among respondents
RDA	Redundancy analysis	Statistical method analyzing the relationship between multiple response and explanatory variables
LM	Linear model	Mathematical method describing a continuous response variable as a function of one or more predictor variables
*R*	Spearman’s correlation	Coefficient describing the strength of a link between two sets of data
*R* ^2^	Coefficient of determination	Descriptive measure of goodness-of-fit for linear models, indicating the percentage of the variance in the dependent variable that is explained by the independent variable(s)

## Data Availability

The data associated with this research are available in a publicly accessible repository at DOI: 10.17632/bgcrby3wbh.1.

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
