# Peer review of "The Ultimate List of the Most Frightening and Disgusting Animals: Negative Emotions Elicited by Animals in Central European Respondents"

_animals, 2021, doi:10.3390/ani11030747_

Round 1

Reviewer 1 Report

Comments on manuscript Ref: Animals: " Negative emotions elicited by animals: The ultimate list of the most frightening and disgusting animals".

This paper focused on negative emotions (fear/disgust) evoked by animal species and aimed to find the most fear- and disgusting-eliciting species. The choice of species was based on previous studies and included vertebrates and invertebrates. The most fear-eliciting species are carnivores, large ungulates sharks and crocodiles, but also to some extend snakes and spider. The most-fear-eliciting animals were endo- and ecto-parasites. This study was based on a questionnaire that was designed to rank the level of fear or disgust elicited by standard photographs of live animals. The methods was rigorous and the data cleanly analysed. I must say that I really enjoyed their study as the paper is particularly well written and easy to follow.

I do not have major comments and have no arguments to question the validity of their results. So, to my mind the paper should be accepted as it is.

My main comment would be that the study involved educated Czech citizen living in the 21st century and therefore reflects the response of people who have access to knowledge especially through international media and reports. This is interesting in a way as the most-fear-eliciting animals are not necessarily the most dangerous in terms of casualties, and are not part of the local fauna. And in particular, the shark for Czeck citizen. And truly dangerous animals are not necessary raked as fear-eliciting species (e.g. the mosquito).  While size and visible teeth are key features, we do not know from this study if this reflects a general unlearned response (due to evolutionary history) or the outcome of our western societies who love sensationalism. Could the outcome have been different if the authors addressed the same questions to populations who have less access to such media? Or is there a universal human mental template of what a fearful animal is? Independent from size…For instance, could fear be elicited by animals that have huge teeth (or claws) regardless of their real size? I think of deep sea fish such as the humpback angler fish, sloane’s viperfish, stoplight loosejaw… They look quite scary to me! In this case, the presence of feature that are perceived as weapons, and potentially deadly, could be the first explanation. And in the discussion (line 380-390), it should be cited first not last. The evolutionary experience with an important predator of human/human ancestor may appear true but does to concern the average Czech citizen.

The same my apply to disgust-eliciting animals. Is there a universal frame related to the strangeness of the morphotype. How are dust mites perceived by people when enlarged? What are the effects of cultural habits? Amongst slimy animals, some have become a delicacy even in Europe (oysters and snails), while other ethnic groups may consume spiders, scorpions and bats.

Anyway, this dees not change the paper, and it is indeed important to understand what triggers negative emotions as they may play a key role in conservation effort. People will be more willing to save the panda or the koala than any species of insect.

Abstract

Line 30: according to

Introduction

Line 119: Four main aims but only three listed below

Results

Line 263: “For disgust, 154 out of 168 comparisons were significant with no disgust-evoking animals ranking lower than non-disgust-evoking animals”. Unclear, please rewrite…

Reviewer 2 Report

In general a well explained and defined job. The main problem I see is that everything is based on a survey of a number of people from a very similar social context. This I think can be confusing. If carried out in the future, a work with similar characteristics would advise a larger sample number and above all more representative. I attach some annotations in order to improve the work:

General comments: It would be interesting to make an index of abbreviations, it would serve to better understand the text.

L27. “scariest” and “most disgusting”  is subjective, I would put “usually considered as..”

L32. What is “W”?

L59. What is the scientific/biology basis for these sensations? I think the introduction should be enriched in this regard. Is there a scientific basis for a relationship between these sensations and some evolutionary adaptation?

L112. In this line you explain properly why you choose these animals (no as in L 27)

L116. Why only one distinct negative emotion?

L179. do you think the chosen population is representative? Do you think that if it had been a more extensive study, the results would have been the same?

L190. since the experiment is based mainly on graphic elements and a test could put images of it in annexes.

L218. If you are interested in the relationship between the different animal groups, it would be interesting to do a statistical study.

L293 Units of Disgusting Ranking? I suppose it is explained in the interview.

L292. Why this animals are considered risk for the health? I think that it should be explained.  

Some groups of animals are superorder, others class others, however, they are a group of several of them (such as big-snake and Red Panda), I think the criteria for grouping animals should be unified, and if this is interesting, do a statistical contrast to detect these differences.
